# EIF4A3-Mediated circ_0008126 Inhibits the Progression and Metastasis of Gastric Cancer by Modulating the APC/β-Catenin Pathway

**DOI:** 10.3390/cancers17020253

**Published:** 2025-01-14

**Authors:** Zeen Wang, Wenxing Chen, Ziwei Wang, Xinglong Dai

**Affiliations:** Gastrointestinal Surgical Unit, The First Affiliated Hospital of Chongqing Medical University, Chongqing 400016, China; wangzeen0611@163.com (Z.W.); chenwenxing0705@163.com (W.C.)

**Keywords:** circ_0008126, miR-502-5p, EIF4A3, APC, gastric cancer

## Abstract

Gastric cancer is one of the common malignant tumors with limited therapeutic targets. CircRNAs, which are endogenous circular RNAs with a closed-loop structure, have been reported to be key regulators and promising biomarkers for various tumors. However, whether and how circ_0008126 is involved in tumor progression still needs to be validated. This study found that low expression of circ_0008126 was associated with increased lymphatic metastasis, later TNM staging, and poorer survival in patients with gastric cancer. Then, the potential biological function of circ_0008126 and the upstream molecular mechanism by which EIF4A3 regulates the formation of circ_0008126 were explored. It was also verified that circ_0008126 inhibited gastric cancer proliferation and metastasis in a miR-502-5p and EIF4A3-dependent manner by attenuating the APC/β-catenin pathway, which extends our understanding of the dual regulatory pathway of circRNA. It provides new insights into the pathogenesis of GC and brings new potential targets for clinical gastric cancer treatment. It provides new insights into the pathogenesis of GC and new potential targets for clinical gastric cancer treatment.

## 1. Introduction

Statistics in 2022 reported that gastric cancer (GC) is ranked as the fifth most common cancerous disease worldwide and the fifth most common primary cause of cancer deaths. East Asia has the highest number of people suffering from CC globally [1]. Risk factors for non-cardia GC are advanced age, alcohol consumption, cigarette smoking, familial predisposition, H pylori, high salted food intake, and previous gastric surgery, which might synergistically contribute to the tumorigenesis and development of GC [2]. Although the treatment for GC includes chemotherapy, immunotherapy, surgery, and target therapy, the metastasis and recurrence of GC in the middle and late stages lead to the failure of the final treatment [3]. In clinical practice, rapid tumor proliferation, direct invasion, and multiple metastatic pathways are common factors leading to poorer prognostic survival in GC patients. The molecular mechanisms underlying the abovementioned malignant biological behaviors of GC are unknown to date. Thus, it is imperative to explore effective biomarkers and therapeutic or diagnostic targets to inhibit and reduce the metastasis and recurrence of GC.

Circular RNAs (circRNAs) are a category of non-coding RNAs with closed-loop structures generated and formed by the reverse splicing of parental genes. By lacking a 5′ cap and 3′ polyadenylated tail, it is highly stable and not easily digested by nucleases. CircRNA dysregulation is compactly correlated to the biological processes of various tumors, containing cell invasion, proliferation, drug resistance, and angiogenesis [4,5]. Circular RNAs have multiple molecular regulatory mechanisms, including serving as protein scaffolds, transcriptional regulators, translating into proteins or polypeptides, and regulating molecules by interacting with miRNAs [6,7]. In gastrointestinal tumors, hsa_circRNA_0000467 played a role in the progression of colorectal cancer by facilitating eIF4A3-mediated c-Myc translation [8]. The aberrantly expressed circRNAs, such as circHIPK3 [9] and circPAK2 [10], are inextricably linked to tumor progression. However, the functions of most circular RNAs have not yet been elucidated. Additionally, EIF4A3 acts as a primary protein of the exon junction complex (EJC), regulating the mRNA splicing and affecting the transport, translation, and mRNA stability [11]. For example, EIF4A3 binds to circZFAND6 pre-mRNA and promotes circZFAND6 biogenesis in breast cancer [12]. A study showed that circKIF4A could interact with EIF4A3 to stabilize SDC1 mRNA, which activates the c-src/FAK pathways and promotes breast cancer progression [13]. However, the role of EIF4A3 in circRNA biogenesis and its function in GC remains unexplored.

This study uncovered that circ_0008126 was prominently down-regulated in GC and then explored that it correlated with increased lymphatic metastasis, advanced TNM stage, and worse survival period in GC patients. EIF4A3 mediated the biosynthesis of circ_0008126, and circ_0008126 could bind to miR-502-5p to regulate APC expression and affect the inactivation of β-catenin signaling. Circ_0008126 could also bind to EIF4A3 in the cytoplasm to increase APC stability and expression. This study demonstrated that EIF4A3-mediated circ_0008126 restrained GC proliferation and metastasis in a miR-502-5p- and EIF4A3-dependent manner by enhancing APC stability and expression, which extends our understanding of dual regulatory pathways of circRNA.

## 2. Materials and Methods

### 2.1. Sample Collection and Patient Data

The 35 cases of pathologically diagnosed GC tissues and matched normal adjacent tissues (5 cm apart) were obtained from March to December 2020 from the First Hospital of Chongqing Medical University. GC patients did not receive immunotherapy, targeted therapy, or chemotherapy before the surgical operation. The tissue samples were frozen and preserved in liquid nitrogen for use and preservation. The clinical data and histopathology of the GC patients were obtained from medical records and pathology reports. All patients voluntarily filled out the informed consent form. This study and collection of tissue samples were authorized and agreed on by the Ethics Committee of the First Affiliated Hospital of Chongqing Medical University.

### 2.2. Cell Culture and Transfection

GES-1, BGC-823, MGC-803, MKN-28, MKN-45, and AGS were incubated and cultured in RPMI 1640 medium (GIBCO, Waltham, MA, USA) mixed with 10% qualified fetal bovine serum (FBS, VivaCell, Denzlingen, Germany) in 5% CO_2_ at 37 °C. These cells were obtained and purchased from the Cell Bank of the Chinese Academy of Sciences (Beijing, China). The 170 bp cDNA of circ_0008126 into LV5 lentivirus was designed and synthesized to construct the circ_0008126 overexpression vector (GenePharma, Shanghai, China). Additionally, the overexpression of the EIF4A3 vector and negative control were constructed and purchased from GenePharma. Then, the GC cells were incubated with 2 μg/mL puromycin, and stably transfected cell lines were obtained. The circ_0008126 and EIF4A3 interfering siRNAs were designed and synthesized from GenePharma. The miR-502-5p inhibitors, miR-502-5p mimics, and negative control (inhibitors NC or mimics NC) were synthesized and purchased from RiboBio (Guangzhou, China). All oligonucleotides were incubated and treated GC cells using Lipofectamine 3000 (Invitrogen, ThermoFisher, Shanghai, China), as detailed in the instructions.

### 2.3. Quantitative Reverse Transcription-PCR (qRT-PCR)

RNAs from human GC tissues and cells were isolated and extracted using TRIzol reagent (Invitrogen, ThermoFisher, China), as detailed in the manufacturer’s instructions. For circRNAs and mRNAs, the treated RNAs were performed for reverse transcription into cDNAs utilizing the PrimeScript RT Reagent Kit (Takara, Kusatsu, Japan). TB Green Premix Ex Taq II was employed to complete the cDNA amplification according to the kit’s protocols (Takara, Japan). The riboSCRIPT Reverse Transcription Kit (Ribobio, China) was utilized to synthesize and reverse transcribed miRNAs, and cDNAs were amplified using SYBR Green qPCR Master Mix (MedChemExpress, Monmouth Junction, NJ, USA). The circ_0008126 primers designed at the circularization site were synthesized and purchased from Geneseed (Guangzhou, China). The other primers (EIF4A3, APC, E-cadherin, Vimentin, PCNA, β-catenin, miR-502-5p, U6, and GAPDH) were synthesized and purchased from Ribobio (Guangzhou, China) or Sangon Biotech (Shanghai, China). The primers are listed and shown in Appendix A. The average amplified values of every molecule were repeated three times independently. The 2^−ΔΔCT^ method was utilized to normalize the molecules to internal control levels.

### 2.4. RNA Fluorescence In Situ Hybridization (FISH)

The specific probes for Cy3-labeled circ_0008126 and FITC-labeled miR-502-5p were arranged and applied for RNA FISH assays (GenePharma, Shanghai, China). The probe sequences of circ_0008126 and miR-502-5p are shown in Appendix A. Slides with GC cells were fixed with 4% paraformaldehyde and mixed with circ_0008126 probe only or circ_0008126 + miR-502-5p probes overnight. Nuclei were counterstained with DAPI. The FISH assay of GC tissues using the circ_0008126 probe was performed similarly to the above method. Fluorescence signal images were observed and photographed using a fluorescence microscope (LSM800, Carl Zeiss AG, Jena, Germany).

### 2.5. RNase R Digestion and Actinomycin D Assay

Total RNA (2 μg) was treated and incubated with 5 U/μg RNase R (Epicentre Technologies, Madison, WI, USA) for 30 min at 37 °C. The levels of circ_0008126 and STX12 were observed and measured using qRT-PCR. GC cells were seeded and cultured in six-well plates (3 × 10^5^ cells/per well), and blocking status was assayed with 2 µg/mL Actinomycin D (Sigma-Aldrich, Burlington, MA, USA) at 4, 8, 12, and 24 h. The RNAs were collected and measured using qRT-PCR at the above times. The expression levels of circ_0008126, STX12, and APC were standardized to the measurements in 0 h.

### 2.6. Cell Counting Kit-8 (CCK-8)

CCK-8 reagent (MedChemExpress, Monmouth Junction, NJ, USA) was purchased and applied to detect GC cell proliferation levels. Briefly, the treated GC cells were cultured at a density of 2000 cells per well using the RPMI 1640 medium incubated at 37 °C for 24 h. After 0, 24, 48, 72, and 96 h later, 10 µL of CCK-8 reagent was added to the medium and incubated for about 1 h. Then, the Microplate Reader was employed to detect the absorbance value for 450 nm.

### 2.7. Colony Formation Assay

The treated GC cells were cultured at a density of 800 per well in 12-well plates. After 2 weeks of culture, the proliferating colonies were washed three times, then fixed with 4% paraformaldehyde for at least 15 min. The colonies were stained using 0.1% Crystal Violet solution for 15 min at room temperature. Lastly, the visible cell colonies were captured, photographed, and counted.

### 2.8. 5-Ethynyl-2′-Deoxyuridine (EdU)

EdU DNA Cell Proliferation Kit (Biotek, Beijing, China) was purchased and employed to detect cellular DNA replication proliferation activity. In brief, the treated GC cells were incubated at a density of 1 × 10^4^ cells in 96-well plates, and the medium was mixed with fresh RPMI 1640 containing 10 μm EdU for another 2 h at 37 °C, with 5% CO_2_. The 4% paraformaldehyde was utilized to fix the GC cells, then treated with 0.5% TritonX-100 and Apollo dye solution separately. The cell nuclei were stained using Hoechst 33,342 dye solution. The staining for GC cells was assessed and photographed by using an inverted fluorescence microscope (LSM800, Carl Zeiss AG, Germany).

### 2.9. Wound Healing Assay

GC cells were cultured at 5 × 10^5^ cells per well in six-well plates. A scratch of the same width was made with a 200 µL pipette when the GC cells had grown to 80% confluence, and images of the wounds were photographed using a microscope at 0 h. Images were retaken 48 h after the scratch at the same location for comparison. The rates of wound closure were calculated and compared.

### 2.10. Transwell Assay

The invasion ability was validated using the Matrigel matrix-coated 24-well Transwell chambers (Jet BioFil, Guangzhou, China). The chamber cavity was filled with 200 µL of serum-free 1640 medium containing 5 × 10^4^ cells in suspension, and about 500 µL medium containing 10% FBS was added to the lower chambers. After 24 h of incubation at 37 °C, the invaded lower membrane cells were washed, and 4% paraformaldehyde was employed to fix the lower chambers for about 20 min. Then, the GC cells in the lower chambers were dyed with 0.1% Crystal Violet solution, and the invaded lower chambers were collected and photographed using a microscope.

### 2.11. Western Blotting (WB)

The total protein of the treated GC cells was extracted and isolated using the RIPA lysis containing complete phosphatase as well as protease inhibitors (Beyotime, Shanghai, China). The cytoplasmic and nuclear proteins were separately collected and extracted using the Nuclear and Cytoplasmic Protein Extraction Kits (Beyotime, China). The BCA kits were utilized to adjust the protein concentration similarly (Beyotime, China). A 10% SDS-PAGE was employed to separate the protein on the gel, and then the protein was transferred to PVDF membranes (Millipore, Billerica, MA, USA). The proteins of the membranes were blocked using 5% non-fat milk powder for 1 h at room temperature and incubated with the corresponding antibody overnight at 4 °C. The above membranes were washed with TBST solution and mixed with the secondary antibody at room temperature for 1 h. Then, bands of samples were visualized and developed using the Ultrasensitive ECL WB Substrate (ZEN BIO, Chengdu, China). The primary antibodies were E-cadherin, Vimentin, APC, Ki-67, PCNA, β-catenin, EIF4A3, and β-actin, which were obtained from the Proteintech (Wuhan, China) or the Abcam (Cambridge, United Kingdom).

### 2.12. Dual-Luciferase Reporter Assay

Circ_0008126 mutant-type or wild-type and APC mutant-type or wild-type luciferase reporter vectors psiCHECK2 that bind to miR-502-5p were synthesized and purchased from Geneseed (China) and called circ_0008126-MUT, circ_0008126-WT, APC-MUT, and APC-WT, respectively. Moreover, the psiCHECK2 APC 3′-UTR reporter vector was co-transfected with EIF4A3 overexpression /siRNA or circ_0008126 overexpression plasmid to determine the 3′-UTR activity of APC. The GC cells were seeded into 24-well plates (5 × 10 cells per well) and co-transfected with circ_0008126, APC reporter vector mixtures, and miR-502-5p mimics or EIF4A3 overexpression/siRNA by lipofectamine 3000 (Invitrogen, China). Two days later, luciferase values were measured and detected using Dual-Luciferase Kits (Promega, Madison, WI, USA) based on the instructions.

### 2.13. Immunofluorescence (IF)

The GC cells overexpressing circ_0008126 or transfected with siRNA were grown and plated on 24-well plates. The 4% paraformaldehyde served to fix the cells on slides for about 20 min. The 0.5% Triton X-100 was employed to permeabilize the fixed cells for 30 min, and they were washed thrice. The washed cells were blocked with 5% bovine serum albumin at room temperature for 1 h. The specific primary antibodies, including EIF4A3, APC, E-cadherin, and Vimentin, were incubated and gained at 4 °C overnight. The fluorescent secondary antibody was obtained from Invitrogen (China) and stained for 1 h at 37 °C the next day. The cell nuclei were adopted to stain with DAPI for 10 min. The images of immunofluorescence staining were recorded and photographed using a confocal fluorescence microscope (LSM800, Carl Zeiss, Germany).

### 2.14. RNA Pull-Down

The biotinylated circ_0008126 probe designed to cross the junction region of circ_0008126 was synthesized and purchased from GenePharma (China). Briefly, streptavidin magnetic beads (BerSinBIO, China) were used to incubate with circ_0008126 probe (3 μg) at room temperature for about 2 h to form circ_0008126 probe-coated magnetic beads. The GC cell lysates were mixed and cultured with the circ_0008126 probe-coated magnetic beads overnight. Subsequently, the washed magnetic beads were purified, including the miR-502-5p or EIF4A3 bound to the mixture, and then the compound was analyzed and extracted using qRT-PCR and WB. Appendix A lists the biotinylated probe sequences for circ_0008126.

### 2.15. RNA Immunoprecipitation (RIP)

RIP was completed in the GC cells using the RIP kit (BerSinBIO, Guangzhou, China), as detailed in the instructions. In brief, the anti-EIF4A3 (Proteintech, China) antibody or IgG-NC antibody was added and incubated with magnetic beads. The GC cell lysates were mixed and cultured with antibody-coated magnetic beads. The magnetic beads were washed, and the precipitated proteins were detected and analyzed for qRT-PCR. Then, we constructed the pcDNA3.1-6 MS2 plasmid (A1, A2, A3, and A4), which embedded the circ_0008126 and its downstream or upstream sequences to transfect into the GC cells. Cell lysates were harvested and pre-precipitated with 50 μL protein A/G-agarose for 2 h (Santa Cruz, CA, USA). The lysates were incubated with the EIF4A3 antibody with rotation at 4 °C, and the immunoprecipitate was obtained for subsequent analysis.

### 2.16. Immunohistochemistry (IHC)

Paraffin-embedded specimens were cut into 5 μm sections, used partly for IHC and partly for hematoxylin and eosin (H&E) staining. Xylene was used to deparaffinize the tissue sections, and then sections were dehydrated using different grades of alcohol. Tissue sections were handled with hydrogen peroxide, and antigenic repair was performed. These sections were mixed and incubated with accordingly specific primary corresponding antibodies overnight at 4 °C, including APC, E-cadherin, Vimentin, PCNA, and Ki-67 (Proteintech, Wuhan, China). Then, the sections were incubated with the corresponding secondary antibody for 1 h and stained with 3′-diaminobenzidine tetrahydrochloride (DAB) and hematoxylin. The marked positive tissues were captured and calculated using a fluorescence microscope (Axio Imager A2, Carl Zeiss, Germany) and Image Pro Plus 6.0.

### 2.17. Animal Model Experiments

The female BALB/c nude mice (4 weeks old) were selected and subcutaneously inoculated with MGC-803 cells transfected OE-circ_0008126 or OE-NC at a density of 1 × 10^7^/100 µL. The tumor size and volume were measured with a caliper weekly (Volume formula: length × width^2^ × 0.5). Subcutaneous tumors were dissected for further qRT-PCR and IHC analysis. Moreover, MGC-803 cells transfected with circ_0008126 overexpressing or OE-NC (5 × 10^6^, 150 µL) were injected into the tail vein of nude mice. Likewise, the livers and lungs of nude mice were dissected and observed. The tumor size and number of nodules were observed and calculated, and metastatic tumor nodules were viewed by HE staining. All animal experimental operations and contents were carried out based on the guidelines and approved by the Animal Care and Use Committee of Chongqing Medical University.

### 2.18. Statistical Analysis

Statistical analyses of all experiments were carried out and calculated using SPSS 24.0 or GraphPad Prism 8.2.1 software, and each experiment was conducted in triplicate. The *p*-values were calculated and performed using the chi-square (χ^2^), student’s *t*-test, or ANOVA. The quantitative data in the graphs are expressed and counted as mean ± SD. Survival rates of follow-up GC patients were statistically analyzed using the Kaplan–Meier method and the log-rank test. The relationship between circ_0008126 levels and clinical variables in GC patients was analyzed using the chi-square (χ^2^) or Fisher’s exact test. Pearson’s correlation coefficient analysis was utilized to analyze the correlations between circ_0008126 and miR-502-5p expression. *p*-values < 0.05 were regarded as the statistical difference, and *p* < 0.01 was viewed as highly significant.

## 3. Results

### 3.1. circ_0008126 Is Verified and Characterized in GC Samples

The clustering heatmap displayed that 3988 circRNAs with >2-fold differential expression were screened for dysregulated expression from the circRNA microarray (*p* < 0.05, Figure 1A). The 10 most differentially expressed circRNAs were detected, and qRT-PCR manifested that circ_0008126 was lower expressed in GC tissues than in paired normal gastric tissues, which was finally selected (Figure 1B,C). The levels of circ_0008126 were markedly decreased in GC cell lines, with the highest expression in AGS cells and the lowest in MGC-803 cells (Figure 1D). Tissue FISH indicated that the circ_0008126 expression level was lower in GC tissues than in paracancerous tissues (Figure 1E). Sanger sequencing confirmed that circ_0008126 existed in GC and originated from the STX12 gene spliced from exon 2 and exon 3, located on chromosome 1 (Figure 1F). Furthermore, STX12 levels did not differ between GC and noncancerous tissues and in different GC stages according to TCGA data (Figure 1G and Appendix A). The expression of circ_0008126 was markedly decreased in lymph node-positive GC patients compared with lymph node-negative GC patients. The expression levels of circ_0008126 in stage III GC patients were less than in stage I GC patients (Appendix A). More importantly, the survival analysis showed that GC patients with a low expression of circ_0008126 had shorter survival times than those with a high expression of circ_0008126 (Figure 1H). Further, circ_0008126 was resistant to RNase R treatment, and degradation was reduced compared to linear STX12 in GC cells (Figure 1I,J). The natural degradation rate of circ_0008126 was also reduced and lower than that of linear STX12 after actinomycin D treatment, indicating that circ_0008126 is not susceptible to degradation and digestion relative to the linear parental gene STX12. (Figure 1K,L). The CSCD database suggested that circ_0008126 is located and expressed in the cytoplasm and nucleus (Appendix A) [14]. Cell FISH further detected that circ_0008126 was primarily distributed in the cytoplasm of GC cells (Figure 1M). The above data describe that circ_0008126 was lowly expressed in GC and might be correlated with the post-transcriptional role of GC.

### 3.2. EIF4A3-Induced circ_0008126 Suppresses the Proliferation, Metastasis, and EMT Ability of GC Cells

The literature has reported that circRNAs are adjusted and controlled by many RBPs, such as DHX9, QKI, ADAR1, and EIF4A3 [15,16]. Thus, the CircInteractome database predicted that EIF4A3 had four sites matching the circ_0008126 precursor RNA flanking sequence (Figure 2A and Appendix A). RIP analysis demonstrated that the circ_0008126 flanking region sequence was enriched at the anti-EIF4A3 group in contrast with the anti-IgG group in GC cells (Figure 2B). The MS2 RIP assays identified the fact that EIF4A3 could bind to both the up- and down-stream sequences of circ_0008126 (Figure 2C and Appendix A). The qRT-PCR assays demonstrated that the overexpression of EIF4A3 facilitated the expression and formation of circ_0008126 in GC cells, with no significant change for STX12 expression. The knockdown of EIF4A3 reduced the expression of circ_0008126 in GC cells without changing the STX12 mRNA level (Figure 2D,E and Appendix A). Based on this, EIF4A3 could mediate circ_0008126 formation ability by binding to the precursor RNA flanking sequence. Then, the MGC-803 cells were transfected with overexpressing circ_0008126, and the AGS cells were transfected with siRNAs targeting circ_0008126 for the function experiments (Figure 2F,G). The CCK-8 and EdU assays exhibited that the proliferative viability of MGC-803 cells in the overexpression circ_0008126 group was observably restrained compared to the OE-NC group (Figure 2H–J), and the silencing of circ_0008126 enhanced the proliferation activity of the AGS cells (Figure 2K–M).

Further, the proliferative clonal activity of GC cells was prominently reduced after the overexpression of circ_0008126 (Figure 3A), and the silencing of circ_0008126 revealed opposite results (Figure 3B). The wound healing assay displayed that the migration ability of MGC-803 cells overexpressing circ_0008126 was weaker than that of the OE-NC group (Figure 3C). In contrast with the Si-NC group, circ_0008126 silencing increased migration ability (Figure 3D). The transwell assays demonstrated that the invasion level of MGC-803 cells overexpressing circ_0008126 was significantly attenuated (Figure 3E). The AGS cells transfected with knocked down circ_0008126 enhanced the invasion level compared with the Si-NC group (Figure 3F). The qRT-PCR and IF assays showed that vimentin expression was downregulated and E-cadherin expression was upregulated after MGC-803 cells overexpressed circ_0008126. After the silencing of circ_0008126 in the AGS cells, vimentin expression was upregulated, and E-cadherin expression was downregulated (Figure 3G–I). These data imply that EIF4A3 mediated the formation of circ_0008126, which affected and suppressed the proliferation, metastasis, and EMT capacity of GC cells.

### 3.3. circ_0008126 Directly Binds to miR-502-5p in GC

CircRNAs can modulate the corresponding molecules mainly by binding to RNA-binding proteins (RBPs) or miRNAs in the cytoplasm. The possible binding miRNAs and RBPs were predicted using the CSCD [14], CircInteractome [17], CircBank [18], and Starbase [19] database tools, which detected four candidate miRNAs and RBPs. In the GC cells, miR-502-5p decreased after circ_0008126 overexpression, while miR-502-5p expression was enhanced when the expression of circ_0008126 was reduced (Figure 4A,B). Thus, circ_0008126 might have a competitive binding relationship with miR-502-5p in GC, so it was a target for the following experiment. Then, the qRT-PCR assays indicated that miR-502-5p showed relatively higher expression in GC cells than in GES-1 cells (Figure 4C). The expression levels of miR-502-5p in 25 pairs of GC tissues exceeded that of normal adjacent tissues (Figure 4D). The miR-502-5p expression levels were inversely inter-related with circ_0008126 expression levels in 25 GC tissues (Figure 4E). Next, the luciferase reporter assays displayed that AGS and MGC-803 cells transfected with miR-502-5p mimics reduced the luciferase activity of circ_0008126-WT, whereas the luciferase activity of circ_0008126-MUT was unchanged (Figure 4F–H). Subsequently, a biotinylated circ_0008126 pull-down assay demonstrated that miR-502-5p was able to be primarily pulled down by circ_0008126 probes in AGS and MGC-803 cells (Figure 4I). The FISH assays demonstrated that miR-502-5p and circ_0008126 were expressed and co-localized in the cytoplasm of MGC-803 and AGS cells (Figure 4J). Additionally, the qRT-PCR assays revealed that the expression levels of circ_0008126 showed a tendency to decrease in the miR-502-5p mimics group compared with the mimics NC group (Figure 4K). The expression of circ_0008126 was more significant in the miR-502-5p inhibitors group than in the NC group (Figure 4L). Meanwhile, the expression level of miR-502-5p in MGC803 cells overexpressing circ_0008126 was prominently less than that in the NC group (Figure 4M). The expression of miR-502-5p in AGS cells knocking down circ_0008126 was slightly exceeded compared with the NC group (Figure 4N). Thus, these data identified that circ_0008126 could function as a miR-502-5p sponge in GC.

### 3.4. circ_0008126 Reduced GC Progression via Targeting miR-502-5p

Rescue experiments were designed to identify whether circ_0008126 affects GC progression by binding to miR-502-5p. The miR-502-5p mimics or mimics-NC were co-transfected with MGC-803 cells overexpressing circ_0008126 or OE-NC for rescue experiments, respectively. Circ_0008126 suppressed the proliferative capacity of MGC-803 cells, as displayed by the CCK-8 and EdU experiments, while miR-502-5p mimics attenuated this suppression (Figure 5A,C,I). We similarly co-transfected miR-502-5p inhibitors and inhibitors-NC with AGS cells circ_0008126 knockdown or Si-NC for further study. The CCK-8 and EdU assays demonstrated that the silencing of circ_0008126 enhanced the growth capacity of AGS cells, whereas the miR-502-5p inhibitors reduced this promotion (Figure 5B,D,J). The wound healing assays demonstrated that the overexpression of circ_0008126 in MGC-803 cells diminished the migratory ability, and the co-transfection of miR-502-5p mimics reduced this inhibitory effect (Figure 5E,K). The silencing of circ_0008126 expression enhanced the migration of AGS cells, whereas co-transfection with miR-502-5p inhibitors reduced this migration-promoting effect (Figure 5F,L). The transwell assays validated the fact that circ_0008126 reduced the invasive ability of MGC-803 cells, and the co-transfection of miR-502-5p mimics weakened this repression effect (Figure 5G,M). The silencing of circ_0008126 expression increased the invasive ability of AGS cells, whereas the co-transfection of miR-502-5p inhibitors attenuated this invasion-promoting ability (Figure 5H,N). These findings indicated that circ_0008126 suppressed GC cell proliferation, migration, and invasion by attenuating the tumor-promoting effects of miR-502-5p.

### 3.5. circ_0008126 Inhibits GC Progression Through miR-502-5p/APC/β-Catenin Pathway

Multiple databases (miRWalk [20], Target Scan [21], miRDB [22], mirDIP [23], and microT-CDS [24]) predicted that various molecules, including APC, were possible target molecules of miR-502-5p (Figure 6A). The DAVID database predicted that the Wnt signaling pathway and gastric cancer were observably enriched (Figure 6B) [25]. Target prediction analysis identified the fact that the 3-UTRs of circ_0008126 and APC shared the same miRNA response elements (MREs) of miR-502-5p (Figure 6C). Then, the qRT-PCR assays indicated that APC expression remarkably increased after the overexpression of circ_0008126 or the inhibition of miR-502-5p in MGC-803 cells, respectively (Figure 6D). APC expression was prominently reduced after AGS cells knocked down circ_0008126 or mixed miR-502-5p mimics, respectively (Figure 6E). The WB and IF assays also exhibited that the overexpression of circ_0008126 upregulated APC expression, and the silencing of circ_0008126 caused a corresponding decrease in APC expression (Figure 6F–H and Appendix A). The luciferase activity of APC-WT distinctly lessened after AGS and MGC-803 cells transfected with miR-502-5p mimics compared to the mimics-NC, while the luciferase activity of APC-MUT was invariant (Figure 6I,J). Next, the WB assays also displayed that MGC-803 cells transfected with the overexpression of circ_0008126 enhanced APC expression levels while decreasing intranuclear β-catenin and total β-catenin expression levels. The silencing of circ_0008126 manifested the opposite trend (Figure 6K and Appendix A). The co-transfection of miR-502-5p mimics in MGC-803 cells overexpressing circ_0008126 resulted in the reduced expression of APC and enhanced expression of intranuclear β-catenin and β-catenin. The co-transfection of miR-502-5p inhibitors in AGS cells with silencing circ_0008126 contributed to increased APC expression and reduced intranuclear β-catenin and β-catenin expression (Figure 6K and Appendix A). Subsequently, TOP/FOP activity was reduced or enhanced after transfection with overexpressing or knockdown circ_0008126 in GC cells, as shown in the TOP/Flash and FOP/Flash assays (Figure 6L,M). The overexpression of circ_0008126 reversed the miR-502-5p mimic-induced gain in TOP/FOP transcriptional activity in MGC-803 cells (Figure 6L). The silencing of circ_0008126 reversed the miR-502-5p inhibitor-induced decrease in TOP/FOP transcriptional activity in AGS cells (Figure 6M). These results demonstrate that circ_0008126 regulated APC expression and attenuated and inhibited β-catenin signaling upon binding to miR-502-5p in GC.

### 3.6. circ_0008126 Interacts with the Cytoplasmic EIF4A3 and Enhances the Stability of APC in GC

Previous studies have confirmed that circRNAs could regulate the expression of their binding RBPs [26]. We further investigated whether circ_0008126 in the cytoplasm binds to RBPs for its function, and the circInteractome database detected multiple matching RBPs for circ_0008126, including EIF4A3 (Appendix A). Given that EIF4A3 is critical for some RNA stability [27,28], we were curious whether cytoplasmic EIF4A3 interacts with circ_0008126 to regulate APC expression. RNA pull-down confirmed that the circ_0008126 probe was able to interact with EIF4A3 in a dose-dependent manner (Figure 7A and Appendix A). Then, the interaction between circ_0008126 and EIF4A3 was tested and confirmed in GC cells using RIP assays (Figure 7B). The FISH-immunofluorescence assays displayed that circ_0008126 co-localized with EIF4A3 in the cytoplasm of MGC-803 and AGS cells (Figure 7C). The qRT-PCR and WB assays explored the effectiveness of overexpressing and knocking down EIF4A3 in GC cells, as shown in Figure 7D,E and Appendix A. The overexpression of EIF4A3 effectively upregulated the mRNA and protein expression of APC (Figure 7F,H and Appendix A); in contrast, the knockdown of EIF4A3 produced the opposite effect (Figure 7G,H and Appendix A). The RNA-protein interaction (RPISeq) website predicted the interaction of EIF4A3 with APC mRNA. Both RF and SVM classifier scores exceeded 0.5, suggesting that EIF4A3 is highly likely to interact with the 3′UTR of APC mRNA (Figure 7I). The RIP assays validated that EIF4A3 could bind to APC mRNA in GC cells (Figure 7J). RNA stability assays revealed that the half-life of APC mRNA was observably prolonged upon the overexpression of EIF4A3, whereas the half-life of APC mRNA was shortened upon EIF4A3 knockdown (Figure 7K,L). In addition, the luciferase reporter assays verified that the ectopic expression of EIF4A3 increased APC 3‘-UTR reporter luciferase activity, whereas the knockdown of EIF4A3 decreased APC 3′-UTR reporter luciferase activity (Figure 7M,N). The overexpression of circ_0008126 reversed the reduction in APC 3′-UTR reporter luciferase activity caused by EIF4A3 knockdown (Figure 7O). The WB analysis showed that the silencing of EIF4A3 decreased the protein level of APC in MGC-803 cells, which was reversed by the ectopic expression of circ_0008126 (Figure 7P and Appendix A). Meanwhile, the overexpression of EIF4A3 increased the protein level of APC in AGS cells, which was reversed by the low expression of circ_0008126 (Figure 7P and Appendix A). These data suggested that circ_0008126 enhanced the mRNA stability of APC via the formation of the circ_0008126/EIF4A3/APC RNA-protein complex.

### 3.7. circ_0008126 Suppresses GC Growth and Metastasis In Vivo

MGC-803 cells transfected with OE-NC or overexpressing circ_0008126 were injected subcutaneously into nude mice. The xenograft tumors were photographed and excised after 4 weeks (Figure 8A). The subcutaneous xenograft mouse model indicated that the xenograft tumors’ weight and volume decreased after circ_0008126 overexpression, suggesting that circ_0008126 could suppress the growth of GC (Figure 8B,C). Then, the collected tumor tissues were tested and verified by IHC experiments, and the results exhibited that after circ_0008126 overexpression, the levels of vimentin, ki-67, and PCNA proteins had been down-regulated, while the levels of APC and E-cadherin proteins had been upregulated (Figure 8D). Similarly, after the overexpression of circ_0008126 by MGC-803 cells, the qRT-PCR assays exhibited that miR-502-5p, PCNA, vimentin, and Ki-67 levels had been down-regulated. Conversely, circ_0008126, APC, and E-cadherin levels were upregulated (Figure 8E). Further, the nude metastasis mouse model revealed that the lung metastatic nodules (Figure 8F,G) and liver metastatic nodules (Figure 8I,J) had observably decreased and lessened after the overexpression of circ_0008126. H&E staining also indicated that the overexpression circ_0008126 had fewer lung and liver metastatic nodules (Figure 8H,K). These results confirmed that circ_0008126 could also suppress GC growth and metastasis in vivo. In summary, EIF4A3-mediated circ_0008126 inhibited gastric cancer proliferation and metastasis in a miR-502-5p- and EIF4A3-dependent manner by attenuating the APC/β-catenin pathway, providing a potential target for the pathogenesis of GC and anti-cancer therapy (Figure 8L).

## 4. Discussion

Many circRNAs have been studied as crucial regulators related to the pathogenesis of multiple cancers and cancer progression [29]. In gastrointestinal tumors, some circRNAs have pro-cancer effects and correlate with the proliferation, invasion, and inhibition of ferroptosis, such as circCOL5A1 [30], and other circRNAs, such as circFGD4, have anti-cancer effects [31]. This study identified an unknown novel circ_0008126 differentially expressed in GC and normal adjacent gastric tissues using and analyzing our group’s circRNA microarray. Circ_0008126 is a circle structure derived from two exons of STX12, which was also down-regulated in GC cells and tissues. The cytoplasmic location of circ_0008126 was found and clarified in GC tissues and cells, indicating that circ_0008126 functions post-transcriptionally. The low expression of circ_0008126 is closely correlated with more lymphatic metastasis, advanced TNM stage, and worse survival of GC patients, suggesting that circ_0008126 is involved in the development and progression of GC.

Then, we noticed that STX12 expression levels in GC tissues were not notably different from those in paracancerous tissues, suggesting that circ_0008126-induced changes in the biological functions of GC are not strongly associated with STX12. Our subsequent studies will further explore the role and value of STX12 in gastric cancer. Recent studies have reported RBPs, such as FUS, DHX9, ADAR1, and EIF4A3, which can regulate the formation and expression of circRNAs [32,33,34]. Our studies showed that EIF4A3 mediated the formation of circ_0008126 by binding to the precursor RNA flanking sequence in GC, as shown in the RIP and qRT-PCR analysis. Further, EIF4A3 could bind to the upstream and downstream sequences of circ_0008126 (MS2 RIP and WB analysis). The overexpression or knockdown of EIF4A3 could increase or decrease the expression levels of circ_0008126 in GC cells. Thus, EIF4A3 in the cytosol mediated the formation of circ_0008126 and promoted its expression in GC, which is at least one of the critical reasons for regulating the expression of circ_0008126. Functionally, circ_0008126 restrained the proliferative, metastasis, and EMT phenotypes of GC in vitro and in vivo. In the downstream mechanism, circRNAs can bind to various miRNAs, thereby suppressing downstream target molecules, highlighting their role as a sponge for miRNAs [35,36]. Subsequently, circ_0008126 interacts with miR-502-5p in GC through RNA pull-down, dual-luciferase reporter and FISH assays. Rescue assays further validated the pro-carcinogenic effect of miR-502-5p in GC and that it could reverse the tumor-suppressing impact of circ_0008126. The above data show that circ_0008126 suppressed GC cell progression by attenuating the tumor-promoting role of miR-502-5p.

Afterward, the possible target molecules of miR-502-5p were enriched for analysis, including the APC and GSK3β. The APC/β-catenin molecules are a signaling pathway that promotes the progression of tumors in the digestive system and is regulated by miRNA through post-transcriptional levels [37,38]. Some studies have demonstrated that APC proteins can restrain the Wnt/β-catenin pathway [39,40]. Our study showed that circ_0008126 and miR-502-5p modulated and impacted the expression level of APC in GC cells, as shown by the dual-luciferase, qRT-PCR, and WB assays. Further, the co-transfection of GC cells with circ_0008126 siRNA1 or circ_0008126 overexpressed and miR-502-5p mimics or inhibitors can reduce or increase cellular TOP/FOP activity, indicating that circ_0008126 and miR-502-5p can regulate the β-catenin pathway, respectively. These findings determine that the circ_0008126/miR-502-5p/APC regulatory network can suppress GC proliferation and metastasis by inhibiting the β-catenin pathway, representing a promising therapeutic intervention for GC.

In addition to functioning as miRNA sponges, there is growing evidence that circRNAs can interact with RBPs and participate in the modulation of tumor-associated genes. For instance, exosomal circ-AHCY recruited EIF4A3 to stabilize TCF4 mRNA and facilitated glioblastoma cell growth via the Wnt signaling pathway [41]. circAURKA suppressed the degradation of CTNNB1 protein by facilitating the interaction of ACLY with CTNNB1 protein, accelerating the growth and metastasis of colorectal cancer [42]. In this study, circ_0008126 was able to interact with EIF4A3 in GC cells, as shown by the RNA pull-down, RIP, and FISH-immunofluorescence assays, suggesting that circ_0008126/EIF4A3 formed an RNA-protein complex. EIF4A3 effectively regulated the expression level of APC in GC. Further, EIF4A3 could bind to APC mRNA and prolong its stability in GC cells, as shown by the RIP, RNA stability, and luciferase reporter assays. The rescue experiments further exhibited that the silencing of EIF4A3 decreased the expression level of APC in GC cells, which was reversed by the overexpression of circ_0008126. Notably, our study showed that circ_0008126 restrains GC proliferation and metastasis in a miR-502-5p- and EIF4A3-dependent manner by enhancing APC stability and expression, which extends our understanding of the dual regulatory pathways of circRNA.

## 5. Conclusions

The above data demonstrate that circ_0008126 is a novel tumor suppressor and inhibits the proliferation, metastasis, and EMT capacity of GC cells. Mechanistically, EIF4A3 mediated circ_0008126 in GC, and circ_0008126 serves as a sponge of miR-502-5p to promote APC expression and attenuate β-catenin signaling. More importantly, circ_0008126 interacts with cytoplasmic EIF4A3 to increase APC mRNA stability and promote its expression. Thus, our findings reveal that circ_0008126 can be regarded as a possible anti-cancer therapeutic target and a potential biomarker for GC.

## Figures and Tables

**Figure 1 cancers-17-00253-f001:**
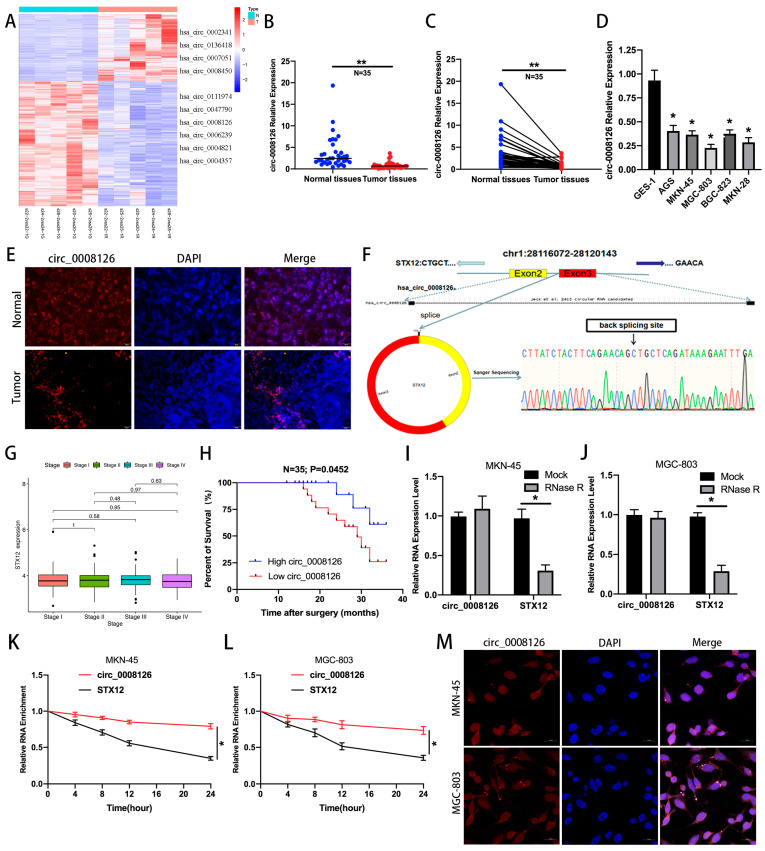
Circ_0008126 is verified and characterized in GC samples. (**A**). The tissue-specific circRNAs in five paired human GC tissues and noncancerous tissues were validated by using a clustered heat map. Hsa_circ_0008126 is marked with a red box. (**B**,**C**). The expression of circ_0008126 in 35 pairs of GC and adjacent tissue samples was detected by using qRT-PCR. (**D**). The expression of circ_0008126 between GC and normal gastric epithelial cell lines was assessed by using qRT-PCR. (**E**). The expression and location of circ_0008126 in GC tissues and adjacent tissues were evaluated by using FISH assays. (**F**). Circ_0008126 was formed by head-to-tail splicing of the 2 and 3 exons of STX12 using Sanger sequencing. (**G**). STX12 levels were validated in different stages of GC tissues from the TCGA database. (**H**). The correlation of circ_0008126 expression levels with the prognosis of GC patients was analyzed using the Kaplan–Meier plotter. (**I**,**J**). The expression of circ_0008126 and STX12 in MKN-45 cells (**I**) and MGC-803 (**J**) was explored by using qRT-PCR after treatment with RNaseR. (**K**,**L**). The expression of circ_0008126 and STX12 at different time points in MKN-45 cells (**K**) and MGC-803 (**L**) was determined by using qRT-PCR after treatment with actinomycin D. (**M**). The expression and location of circ_0008126 in the cytoplasm of GC cells were confirmed by using FISH assays. Values are shown as the mean ± standard error. * *p* < 0.05, ** *p* < 0.01.

**Figure 2 cancers-17-00253-f002:**
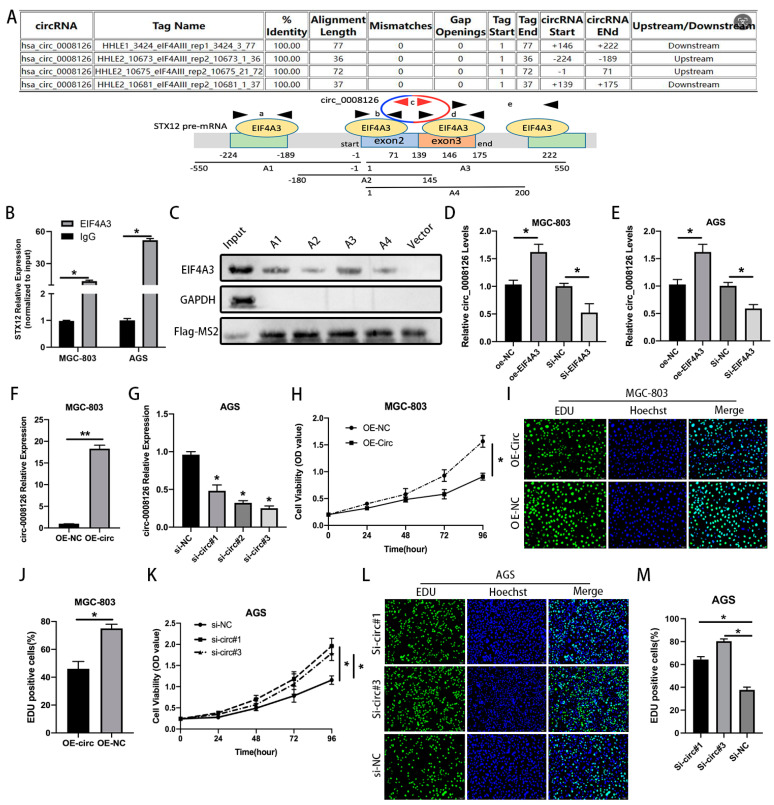
EIF4A3-induced circ_0008126 suppresses the proliferation, metastasis, and EMT ability of GC cells. (**A**). The binding sites for EIF4A3 in the flanking sequences of the STX12 pre-mRNA transcript were predicted using CircInteractome. (**B**). The putative binding sites of EIF4A3 in STX12 pre-mRNA (circ_0008126 precursor RNA flanking region) were demonstrated via the RIP assays. (**C**). Western blot analysis of A1, A2, A3, and A4 protein with MS2-RIP assay. Vector and GAPDH were used as negative controls. (**D**,**E**). The expression of circ_0008126 was detected in MGC-803 cells (**D**) and AGS cells (**E**) after EIF4A3 down-regulation or up-regulation by using qRT-PCR. (**F**). The expression of circ_0008126 in MGC-803 cells stably transfected with LV5 lentiviral-circ_0008126 was detected by using qRT-PCR. (**G**). The expression of circ_0008126 in AGS cells treated with siRNA was examined by using qRT-PCR. (**H**). The proliferation of MGC-803 cells was measured by using CCK-8 assays after the overexpression of circ_0008126. (**I**,**J**). The DNA synthesis of MGC-803 cells was tested via EdU assays after the overexpression of circ_0008126. (**K**). The proliferation of AGS cells was explored using CCK-8 assays after the silencing of circ_0008126. (**L**,**M**). The DNA synthesis of AGS cells was observed using EdU assays after the silencing of circ_0008126. Values are shown as the mean ± standard error. * *p* < 0.05, ** *p* < 0.01.

**Figure 3 cancers-17-00253-f003:**
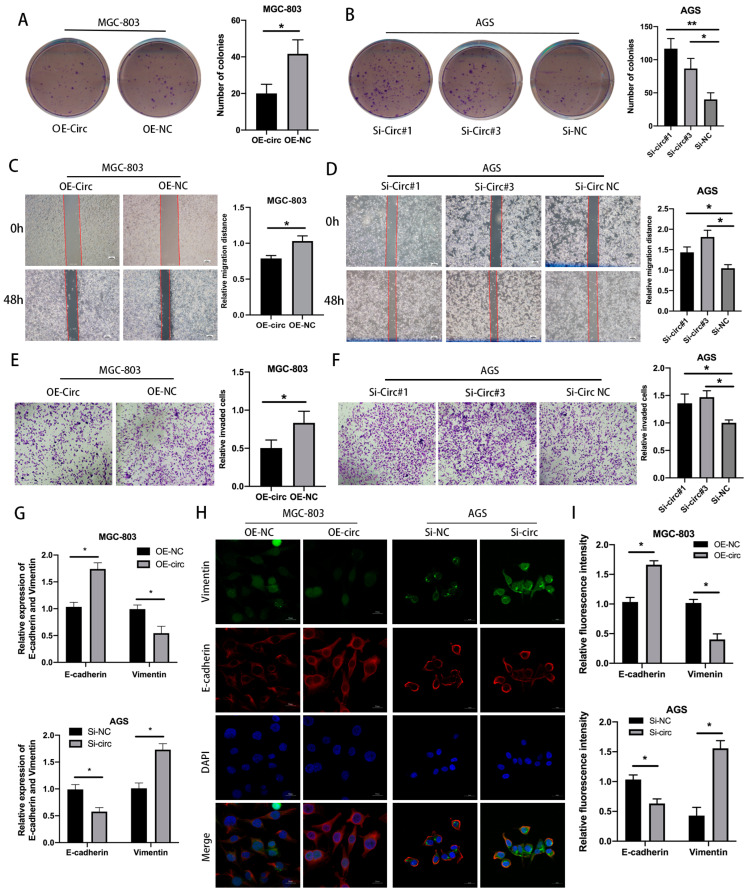
EIF4A3-induced circ_0008126 suppresses the proliferation, metastasis, and EMT ability of GC cells. (**A**,**B**). Colony formation ability was verified in MGC-803 cells after the overexpression (**A**) and knockdown (**B**) of circ_0008126. (**C**,**D**). Wound healing assays examined the migration ability of cells after the overexpression (**C**) and knockdown (**D**) of circ_0008126. (**E**,**F**). Transwell invasion assays verified the invasion ability of cells after the overexpression (**E**) and knockdown (**F**) of circ_0008126. (**G**). The expression of vimentin and E-cadherin after circ_0008126 overexpression or knockdown was validated by qRT-PCR assays. (**H**,**I**). The fluorescence intensities of vimentin and E-cadherin after circ_0008126 overexpression or knockdown were detected by using IF assays. Values are shown as the mean ± standard error. * *p* < 0.05, ** *p* < 0.01.

**Figure 4 cancers-17-00253-f004:**
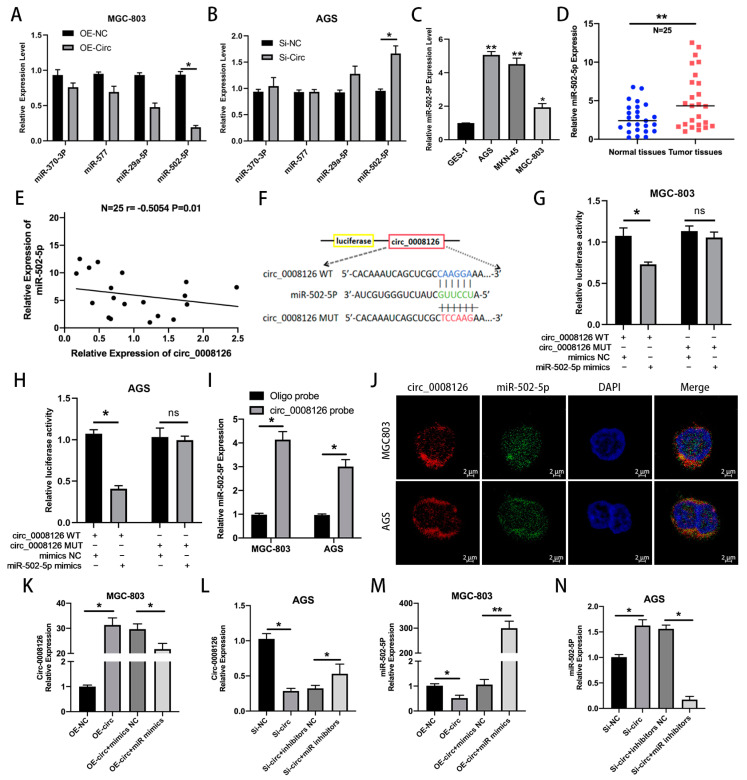
Circ_0008126 directly interacts with miR-502-5p in GC. (**A**,**B**). The expression changes for candidate miRNAs after circ_0008126 overexpression or knockdown were determined using qRT-PCR. (**C**). The expression levels of miR-502-5p in AGS, MGC-803, MKN-45, and GES-1 were examined using qRT-PCR. (**D**) The differential expression of miR-502-5p between 25 paired GC and adjacent tissue samples was detected using qRT-PCR. (**E**). Pearson correlation analysis indicated circ_0008126 and miR-502-5p expression levels in 25 GC tissues. (**F**). Schematic illustration of circ_0008126 -WT and circ_0008126 -MUT luciferase reporter vectors. (**G**,**H**). The relative luciferase activity of the circ_0008126-WT and circ_0008126-MUT were observed in MGC-803 (**G**) and AGS (**H**) cells. (**I**). The expression levels of miR-502-5p in MGC-803 and AGS cells were pulled down by circ_0008126 or oligo probe. (**J**). The co-localization of circ_0008126 and miR-502-5p in GC cells was validated by FISH assays. (**K**,**L**). The expression levels of circ_0008126 in each group were tested using qRT-PCR. MGC-803 cells were transfected with OE-NC, OE-circ, OE-circ + mimics-NC, or OE-circ + miR-502-5p mimics (**K**). AGS cells were transfected with Si-NC, Si-circ, Si-circ+ inhibitors-NC, or Si-circ + miR inhibitors (**L**). (**M**). The expression levels of miR-502-5p in OE-NC, OE-circ, OE-circ+ mimics-NC, or OE-circ + miR-502-5p mimics groups were detected using qRT-PCR. (**N**). The expression levels of miR-502-5p in Si-NC, Si-circ, Si-circ + inhibitors-NC, or Si-circ + miR inhibitors groups were verified using qRT-PCR (**N**). Values are shown as the mean ± standard error. * *p* < 0.05, ** *p* < 0.01.

**Figure 5 cancers-17-00253-f005:**
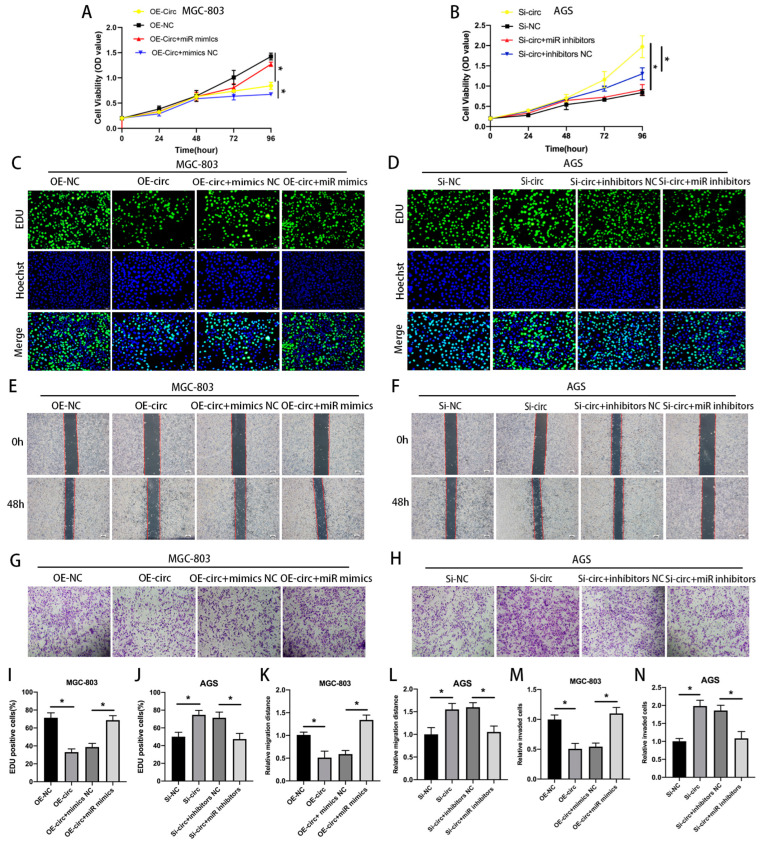
Circ_0008126 reduced GC progression via targeting miR-502-5p. (**A**). The proliferation ability of the OE-NC, OE-circ, OE-circ + mimics-NC, or OE-circ + miR-502-5p mimics groups in MGC-803 cells was measured using CCK-8 assay. (**B**). The proliferation ability of the Si-NC, Si-circ, Si-circ+ inhibitors-NC, or Si-circ + miR inhibitors groups in AGS cells was determined using CCK-8 assays. (**C**,**I**). The DNA synthesis of the OE-NC, OE-circ, OE-circ+ mimics-NC, or OE-circ + miR-502-5p mimics groups in MGC-803 cells were observed using EdU assays. (**D**,**J**). The DNA synthesis of the Si-NC, Si-circ, Si-circ+ inhibitors-NC, or Si-circ + miR inhibitors groups in AGS cells was examined using EdU assays. (**E**,**K**). The migration ability of the OE-NC, OE-circ, OE-circ+ mimics-NC, or OE-circ + miR-502-5p mimics groups in MGC-803 cells were verified using wound healing assays. (**F**,**L**). The migration ability of the Si-NC, Si-circ, Si-circ+ inhibitors-NC, or Si-circ + miR inhibitors groups in AGS cells was tested using wound healing assays. (**G**,**M**). The invasion ability of the OE-NC, OE-circ, OE-circ+ mimics-NC, or OE-circ + miR-502-5p mimics groups in MGC-803 cells were validated using transwell invasion assays. (**H**,**N**). The invasion ability of the Si-NC, Si-circ, Si-circ+ inhibitors-NC, or Si-circ + miR inhibitors groups in AGS cells was confirmed by the transwell invasion assays. Values are shown as the mean ± standard error. * *p* < 0.05.

**Figure 6 cancers-17-00253-f006:**
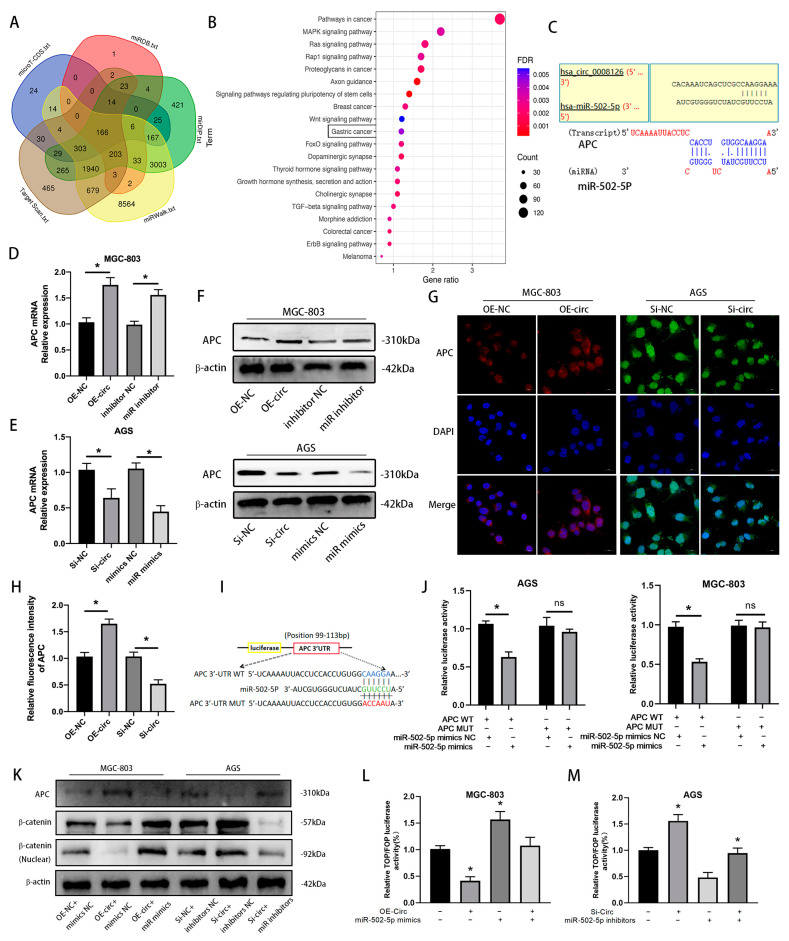
Circ_0008126 inhibits GC progression through miR-502-5p/APC/β-catenin signaling. (**A**). The miRWalk, Target Scan, miRDB, mirDIP, and microT-CDS databases predicted the potential target molecules of miR-502-5p. (**B**). The miR-502-5p predicted target molecules were subjected to KEGG enrichment analysis, and the top 20 signaling pathways with significant enrichment were labeled. (**C**). Schematic representation of the potential binding sites of circ_0008126 to miR-502-5p and miR-502-5p to the 3′UTR of APC. (**D**). The expression levels of APC after overexpression of circ_0008126 and miR-502-5p inhibitors were determined using qRT-PCR. (**E**). The expression levels of APC after the silencing of circ_0008126 and miR-502-5p mimics were detected using qRT-PCR. (**F**). The expression levels of APC after the overexpression or silencing of circ_0008126 were verified using WB. (**G**,**H**). The fluorescence intensity of APC after the overexpression or silencing of circ_0008126 was tested using the IF assays. (**I**). Schematic illustration of APC-WT and APC-MUT luciferase reporter vectors. (**J**). The relative luciferase activity of the APC-WT and APC-MUT were examined in AGS (**I**) and MGC-803 (**J**) cells. (**K**). The expression of APC, β-catenin and intranuclear β-catenin after OE-NC, OE-circ, OE-circ+ mimics-NC, OE-circ + miR-502-5p mimics groups and Si-NC, Si-circ, Si-circ+ inhibitors-NC, or Si-circ + miR inhibitors groups were determined using WB. (**L**,**M**). The dual-luciferase assays showed the effect on TOP/FOP reporter activity in GC cells after transfection with mimics NC, miR-502-5p mimics, inhibitors NC, miR-502-5p inhibitors, circ_0008126 overexpression or siRNA, and miR-502-5p mimics + circ_0008126 overexpression or miR-502-5p inhibitors + siRNA. Values are shown as the mean ± standard error. * *p* < 0.05.

**Figure 7 cancers-17-00253-f007:**
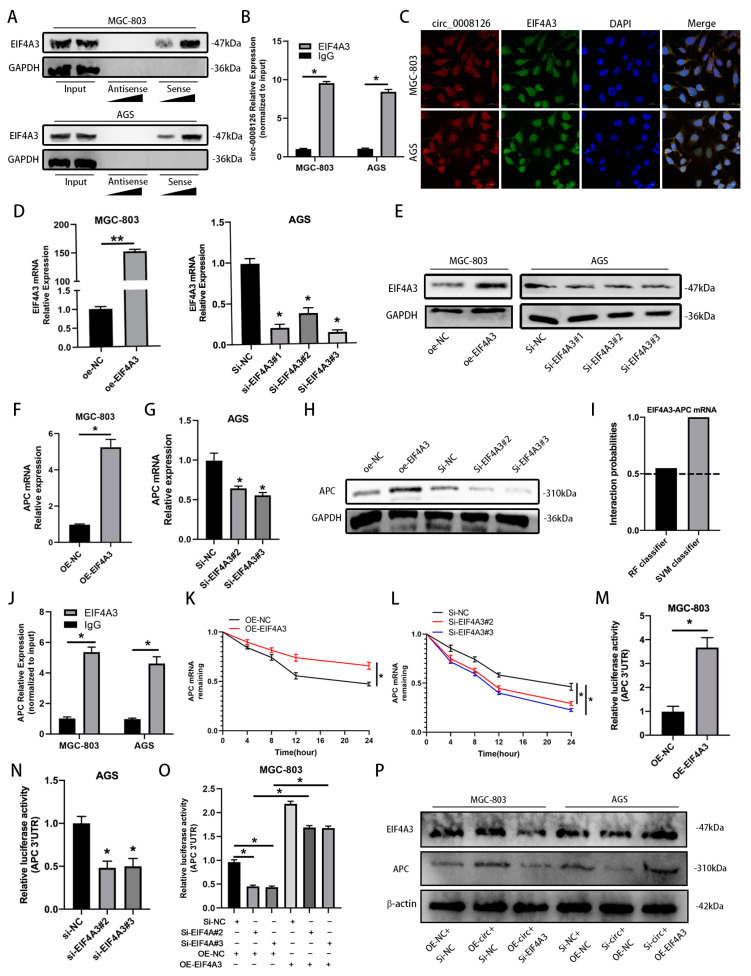
Circ_0008126 interacts with the cytoplasmic EIF4A3 and enhances the stability of APC in GC. (**A**). EIF4A3 protein was pulled down by biotin-labeled circ_0008126 antisense or sense probes, as detected by WB analysis. (**B**). The association of EIF4A3 and circ_0008126 (upper) was shown via RIP assays. IgG was used as the negative control. (**C**). The subcellular location of circ_0008126 and EIF4A3 in GC cells was evaluated using FISH and IF assays. Nuclei were stained with DAPI. (**D**). The expression of EIF4A3 in GC cells transfected with oe-NC or EIF4A3 overexpression or Si-NC or Si-EIF4A3 was validated using qRT-PCR assays. (**E**). The expression of EIF4A3 in GC cells transfected with oe-NC or EIF4A3 overexpression or Si-NC or Si-EIF4A3 was validated using WB analysis. (**F**,**G**). The expression of APC in GC cells transfected with oe-NC or EIF4A3 overexpression or Si-NC or Si-EIF4A3 was verified using qRT-PCR assays. (**H**). The expression of APC in GC cells transfected with oe-NC or EIF4A3 overexpression or Si-NC or Si-EIF4A3 was determined via WB analysis. (**I**). The interaction probabilities of EIF4A3 with the 3′UTR of APC mRNA were predicted by using the RNA-Protein interaction prediction (RPISeq) website. Predictions with probabilities > 0.5 were considered to represent that the corresponding RNA and protein are likely to interact. (**J**). The association of EIF4A3 and APC was tested using RIP assays. IgG was used as the negative control. (**K**,**L**). RNA stability assays revealed the remaining levels of APC mRNA in GC cells transfected with the overexpression of EIF4A3 (**K**) or siRNA specifically targeting EIF4A3 (**L**). (**M**,**N**). The luciferase activities of APC 3′UTR were measured after the overexpression (**M**) or knockdown (**N**) of EIF4A3 in GC cells, respectively. (**O**). The luciferase activities of APC 3′UTR were measured in GC cells transfected with si-NC, si-EIF4A3 #1, or si-EIF4A3 #2, and those co-transfected with OE-NC or OE-circ_0008126. (**P**). The expression of APC in GC cells transfected with si-NC, si-EIF4A3, OE-NC, or OE-EIF4A3 and those co-transfected with si-NC, si-circ_0008126, OE-NC or OE-circ_0008126. Values are shown as the mean ± standard error. * *p* < 0.05, ** *p* < 0.01.

**Figure 8 cancers-17-00253-f008:**
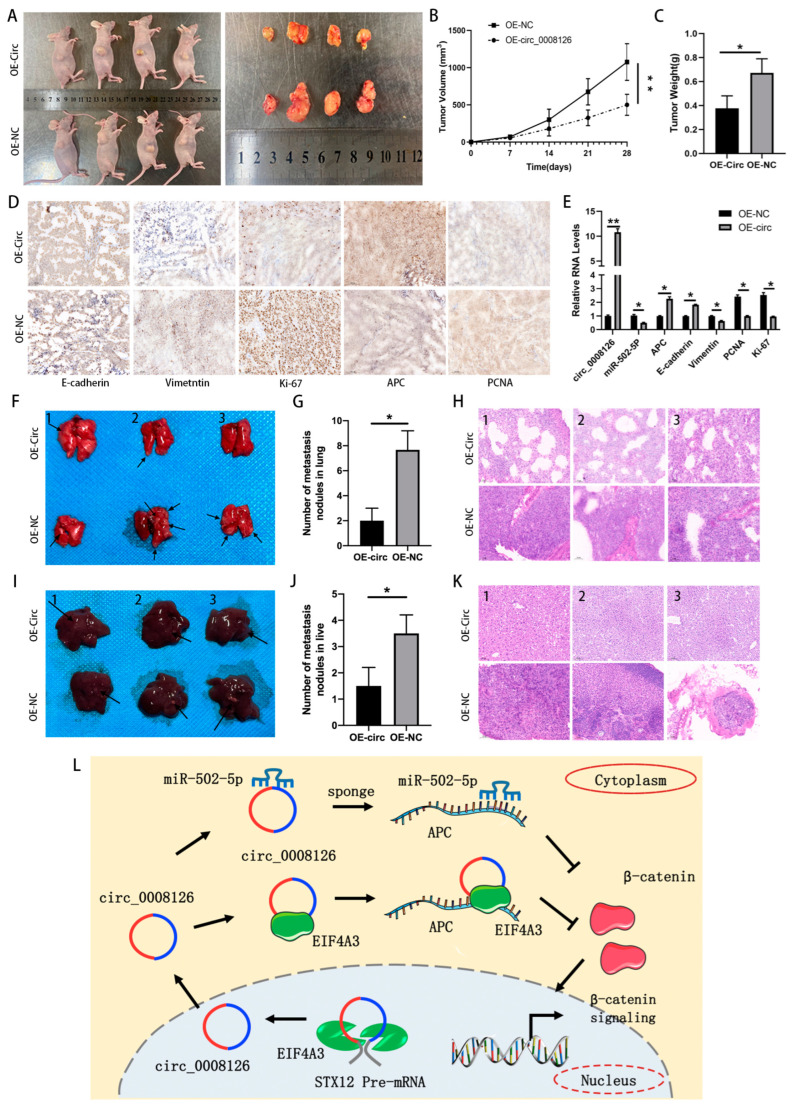
Circ_0008126 suppresses GC tumorigenesis and metastasis in vivo. (**A**). The subcutaneous tumor tissues in the circ_0008126 overexpression and OE-NC groups were excised and photographed. (**B**,**C**). The volume and weight of subcutaneous tumor tissues in the circ_0008126 overexpression and the OE-NC groups were regularly observed. (**D**). The expression levels of E-cadherin, vimentin, Ki-67, APC, and PCNA in the circ_0008126 overexpression and the OE-NC groups were verified by the IHC assays. (**E**). The expression levels of circ_0008126, miR-502-5p, APC, E-cadherin, vimentin, PCNA, and Ki-67 in the circ_0008126 overexpression and the OE-NC groups were detected by qRT-PCR. (**F**,**G**). Representative images of lung metastatic nodules in the circ_0008126 overexpression and the OE-NC groups are indicated by arrows. (**H**). The H&E staining of lung metastatic nodules in the circ_0008126 overexpression and the OE-NC groups was observed. (**I**,**J**). Representative images of liver metastatic nodules in the circ_0008126 overexpression and the OE-NC groups are indicated by arrows. (**K**). The H&E staining of liver metastatic nodules in the circ_0008126 overexpression and the OE-NC groups was observed. (**L**). The schematic diagram describes the mechanism by which EIF4A3-mediated circ_0008126 inhibits the proliferation and metastasis of GC through the regulation of APC by sponging miR-502-5p and interacting with EIF4A3, respectively. Values are shown as the mean ± standard error. * *p* < 0.05, ** *p* < 0.01.

## Data Availability

All data generated or analyzed during this study are contained in this paper or Appendix A. The datasets presented in this study can be found in online repositories. The data will be made available upon request.

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
