# Peer review of "EIF4A3-Mediated circ_0008126 Inhibits the Progression and Metastasis of Gastric Cancer by Modulating the APC/β-Catenin Pathway"

_cancers, 2025, doi:10.3390/cancers17020253_

Round 1

Reviewer 1 Report

Comments and Suggestions for Authors

The manuscript by Zeen Wang et al "EIF4A3-Mediated circ_0008126 Inhibits the Progression and Metastasis of Gastric Cancer by Modulating the APC/β-Catenin Pathway" is dedicated to the thorough study of the almost unknown circular RNA. Authors studied the expression levels and clinical values of circ_0008126 in GC patients and proposed an underlined molecular mechanism. Authors overestimate the role of EIF4A3 in the processes. This RBP is known for multiple functions, including RNA binding in stress granules and nonsense-mediated mRNA decay. Thus, the arm circ_0008126 - miR-502-5p - beta-catenin looks reasonable, while contribution of direct interactions with APC can be minimal or is a result of indirect regulation. Please provide weight of mice (Fig. 8A). Figures 1M, 4J, 6G, 8D, 8H should be increased - they are meaningless now. Nomination of circ_0008126 as a biomarker and a therapeutic target for GC requires additional studies. 

Author Response

3. Point-by-point response to Comments and Suggestions for Authors

Comments 1: The manuscript by Zeen Wang et al "EIF4A3-Mediated circ_0008126 Inhibits the Progression and Metastasis of Gastric Cancer by Modulating the APC/β-Catenin Pathway" is dedicated to the thorough study of the almost unknown circular RNA. Authors studied the expression levels and clinical values of circ_0008126 in GC patients and proposed an underlined molecular mechanism. Authors overestimate the role of EIF4A3 in the processes. This RBP is known for multiple functions, including RNA binding in stress granules and nonsense-mediated mRNA decay. Thus, the arm circ_0008126 - miR-502-5p - beta-catenin looks reasonable, while contribution of direct interactions with APC can be minimal or is a result of indirect regulation. Please provide weight of mice (Fig. 8A). Figures 1M, 4J, 6G, 8D, 8H should be increased - they are meaningless now. Nomination of circ_0008126 as a biomarker and a therapeutic target for GC requires additional studies.

Response 1:

Firstly, we would like to thank the reviewers for their positive comments. EIF4A3 has a variety of regulatory functions, and subsequent research will continue to study the relationship between circRNA and EIF4A3 in GC. Then, we have made the following revisions to address some of the issues. We have provided the weight of mice in Figure 8A, and the four-week female BALB/c nude mice weigh 16-20g. Please see the revised Figure 8A for details (Page 19).

Next, images of the FISH, IHC, and H&E experiments (Fig. 1M, Fig. 4J, Fig. 6G, Fig. 8D, and Fig. 8H) were low resolution. Still, the original images were of good resolution, and we re-uploaded them for combination. Figures 1M and 4J are FISH experiments verifying the localization of circ_0008126 in gastric cancer and the binding relationship between circ_0008126 and miR-502. Figures 6G, 8D, and 8H are the results of IF, IHC, and HE staining, essential in verifying the downstream molecular and phenotypic changes regulated by circ_0008126. Please see the revised Figures for details.

Lastly, the nomination circ_0008126 is a biomarker, and the therapeutic target for GC is too broad and definitive. We searched the full text and moderately reduced the degree of absoluteness, e.g., replacing it with circ_0008126 can be considered a possible anti-cancer therapeutic target and a potential biomarker for GC (Page 21, line 666).

4. Response to Comments on the Quality of English Language

Point 1: Quality of English Language (x) The quality of English does not limit my understanding of the research

Response 1:

      We thank the reviewers for their recognition, and our manuscript was edited and polished by an English-speaking professional organization.

Reviewer 2 Report

Comments and Suggestions for Authors

In this manuscript authors demonstrated that circ_008126 inhibited oncogenic traits both in vitro and in vivo in gastric cancer cells. As a tumor suppressor circ_0018126 mediated by EIF4A3 served as a sponge of miR-502-5p to promote APC expression and attenuate b-catenin signaling. The manuscript was well written, the experiment design did support the main conclusion. Yet there are still some concerns for authors to address before it’s eligible for acceptance. Here are the specific comments.

Concerns:

1.    When authors addressed STX12 levels in GC tissues (line 283-284) Supplementary Figure S1A showed clear differences although there is no statistical analysis on that. Based on the results of Figure 1H, Supplementary Figure S1B and S1C STX12 also has an important role in GC. Authors need to further address that.

2.    Line 292-294, authors used RNase R and actinomycin D treatment to show circ_008126 and STX12 levels. What were the conclusions from these results. Why authors focus on circular RNA instead of STX12?

3.    Line 324- 326, authors interpreted STX12 mRNA level unchanged based on Supplementary Figure S1F and S1G, however the figures showed statistical significance. Either the statement or the figure made a mistake.

4.    In Figure 3G and 3H, RT-qPCR analyses of EMT genes are preferred to confirm EMT capacity.

5.    In Figure 4K and 4M, authors showed miR-502-5P and circ_008126 level in MGC-803. In OE-circ, circ_008126 was 30 folder higher than OE-NC, miR-502-5P was 50% lower. However, in OE-circ plus miR mimics circ_008126 was 30% lower than control while miR-502-5P was about 600 folder higher. Why there were huge differences in these two groups?

6.    In Figure 5I-5N, OE-circ and OE-circ plus mimics NC in MGC-803 cells should have relatively equal proliferation and migration rate but results showed big variation. Similarly, controls in MGC-803 cells should perform same.

7.    Western blots in Fig6F, 6K and Figure 7P did not match the original blots in the supplementary data.

8.    All the immunofluorescent images need quantifications to demonstrate the intensity change.

9.    In Figure 8D, IHC staining panels need better resolution images.

Author Response

3. Point-by-point response to Comments and Suggestions for Authors

Comments 1: In this manuscript authors demonstrated that circ_008126 inhibited oncogenic traits both in vitro and in vivo in gastric cancer cells. As a tumor suppressor circ_0018126 mediated by EIF4A3 served as a sponge of miR-502-5p to promote APC expression and attenuate b-catenin signaling. The manuscript was well written, the experiment design did support the main conclusion. Yet there are still some concerns for authors to address before it’s eligible for acceptance. Here are the specific comments. 1. When authors addressed STX12 levels in GC tissues (line 283-284) Supplementary Figure S1A showed clear differences although there is no statistical analysis on that. Based on the results of Figure 1H, Supplementary Figure S1B and S1C STX12 also has an important role in GC. Authors need to further address that.

Response 1:

We sincerely thank the reviewers for their suggestions; the revisions are much needed. We have made the following revisions to address some of the issues. Firstly, this study focused on the significantly low expression of circular RNA circ_0008126 in gastric cancer and its upstream/downstream molecules. The upstream parental gene of circ_0008126, STX12, was not statistically differently expressed in gastric cancer and paracancerous tissues and different tumor stages. The value of judging the prognosis of gastric cancer patients based on the median value of STX12 expression is not great. Also, the article was not carried out to elaborate on the functional role of STX12. After careful consideration, the survival curves of STX12 can be deleted, and we revised it in the discussion section with a note that subsequent studies will further explore the role and value of STX12 in gastric cancer (Page 20, line 612).

Comments 2: Line 292-294, authors used RNase R and actinomycin D treatment to show circ_008126 and STX12 levels. What were the conclusions from these results. Why authors focus on circular RNA instead of STX12?

Response 2:

According to the Reviewer's comments, RNase R and actinomycin D assays indicated that circ_0008126 is not susceptible to degradation and digestion relative to the linear parental gene STX12. Circ_0008126 can form a ring, is differentially expressed, is structurally stable, and may be a potential biomarker in gastric cancer. Please see the revised on Page 7, line 294.

Circ_0008126 was found to be differentially expressed from gastric cancer circRNA microarrays and is the main molecule in this study, which includes the biosynthesis upstream role of circ_0008126 and the downstream phenotypic and molecular role of circ_0008126 in regulating gastric cancer. Bioinformatics showed no significant differential expression of STX12 in gastric cancer tissues, normal tissues, and different gastric stages. Whether STX12 can significantly regulate the gastric cancer phenotype is uncertain, so we will follow up on the study.

Comments 3: Line 324- 326, authors interpreted STX12 mRNA level unchanged based on Supplementary Figure S1F and S1G, however the figures showed statistical significance. Either the statement or the figure made a mistake.

Response 3:

We thank the reviewers for their questions. The statistical labeling of supplementary figures S1F and S1G was incorrect, and we have corrected it. Please see the revised Supplementary Figures S1D and S1E (The Previous Figures S1F and S1G).

Comments 4: In Figure 3G and 3H, RT-qPCR analyses of EMT genes are preferred to confirm EMT capacity.

Response 4:

According to the Reviewer's comments, we performed qRT-qPCR analysis to find changes in the expression of EMT-related molecules (E-cadherin/Vimentin) after intervention circ_0008126. Please see the revised Figure 3G and its legend, Page 10, line 357.

Comments 5: In Figure 4K and 4M, authors showed miR-502-5P and circ_008126 level in MGC-803. In OE-circ, circ_008126 was 30 folder higher than OE-NC, miR-502-5P was 50% lower. However, in OE-circ plus miR mimics circ_008126 was 30% lower than control while miR-502-5P was about 600 folder higher. Why there were huge differences in these two groups?

Response 5:

Thanks to the Reviewer's question, qRT-PCR experiments showed that after overexpression of circ_0008126 in gastric cancer cells, the overexpression of circ_0008126 was effective and elevated (30 folder higher, Figure 4K), and the expression of miR-502 was slightly decreased (1 folder lessen, Figure 4M). We repeated these experiments in which the expression of miR-502 was indeed significantly increased by almost 300-fold upon the addition of miR-502 mimic to gastric cancer cells overexpressing circ_0008126 and was unaffected by the overexpression of circ_0008126 (Revised Figure 4M). In contrast, the expression of circ_0008126 was decreased (decreased by ten folders, Figure 4K). The qRT-PCR results (Figures 4K, 4M) illustrated that the suppressor effect of circ_0008126 in gastric cancer could be partially reversed by miR-502 mimic. Similarly, the silencing of circ_0008126 in gastric cancer could be partially reversed by miR-502 inhibitor (Figures 4L, 4N).

Comments 6: In Figure 5I-5N, OE-circ and OE-circ plus mimics NC in MGC-803 cells should have relatively equal proliferation and migration rate but results showed big variation. Similarly, controls in MGC-803 cells should perform same.

Response 6:

Thanks again to the reviewers for their suggestions. The revision is necessary. The actual data of EdU (Figures 5C and 5D), Wound healing (Figures 5E and 5F), and Transwell (Figures 5G and 5H) depicted in the bar graphs in Figures 5I-5N have inaccuracies, and we revised them. Please see the revised Figures 5I-5N.

Comments 7: Western blots in Fig6F, 6K and Figure 7P did not match the original blots in the supplementary data.

Response 7:

We thank the reviewers for their careful review. The authors ignored the original blots in the Supplementary Data, and some bands were not clearly labeled (Figures 6F, 6K) and overstretched (Fig. 7P), which we adjusted separately. Please see the revised Figures 6F, 6 K, and 7P. In addition, we re-ran the WB experiments, which yielded similar results, and we can add to the Supplementary Figures if needed.

Comments 8: All the immunofluorescent images need quantifications to demonstrate the intensity change.

Response 8:

We thank the reviewers for their suggestions. Due to the fluorescence intensity of the detected molecules changes after overexpression or silencing of circ_0008126, it is necessary to quantify the fluorescence data. However, in the absence of circ_0008126 interventions, some cellular localization or co-localization FISH experiments are of little significance for the quantification of expression intensity. Therefore, we performed quantitative fluorescence analysis of changes in molecular expression after intervention on circ_0008126 using the ImageJ software. Please see revised Figure 3H and Page 10, Figure 6G and Page 15.

Comments 9: In Figure 8D, IHC staining panels need better resolution images.

Response 9:

The reviewer rightly reminded us that the image clarity was poor. We have ensured the image quality by re-uploading the original high-resolution figure. Please see the revised Figure 8D for details.

4. Response to Comments on the Quality of English Language

Point 1: Quality of English Language (x) The quality of English does not limit my understanding of the research

Response 1:

      We thank the reviewers for their recognition, and our manuscript was edited and polished by an English-speaking professional organization.

Reviewer 3 Report

Comments and Suggestions for Authors

The manuscript is really interesting particularly for the use of two cell lines. However, the authors should make some revision to the text as follows:

- Line 151, page 4, authors should correct "wall" to "well";

- "GC cells" refers to gastric cancer cell line? How did the authors seed the same number of cells for the experiment? Also, it is not clear whether the authors determined the number of cells to treat for each experiment since for CCK-8 and EdU the number is significantly different;

- In figures I and J, RNaseR should be written the same way also for MGC-803 plot;

- In figure 1, E and G should be written in bold. It is the same for figure 2 and the other figures;

- Could it be possible to add uncropped blots as additional files?

- The authors reported so many Western blots but for all of them there is no densitometric analysis. The latter is strictly necessary to calculate the difference in expression level between multiple conditions. Could the authors perform densitometric analysis? How many replicates were performed?

- To make the reading clearer and simpler, I would suggest reporting more experimental information in the results.

Author Response

3. Point-by-point response to Comments and Suggestions for Authors

Comments 1: The manuscript is really interesting particularly for the use of two cell lines. However, the authors should make some revision to the text as follows:

- Line 151, page 4, authors should correct "wall" to "well";

Response 1:

Firstly, we sincerely thank the reviewers for their suggestions. Revisions are much needed. Thanks to the reviewers for finding errors ("wall"); we have revised and see Line 153, Page 4.

Comments 2: - "GC cells" refers to gastric cancer cell line? How did the authors seed the same number of cells for the experiment? Also, it is not clear whether the authors determined the number of cells to treat for each experiment since for CCK-8 and EdU the number is significantly different;

Response 2:

Thanks to the reviewer's question, GC cells refer to gastric cancer cell lines. The number of cells varied for different experiments based on previous experimental experience, but the number of cells in the experimental and control groups was consistent. The number of cells per replicate well can be controlled using a cell counting plate.

Comments 3: - In figures I and J, RNaseR should be written the same way also for MGC-803 plot;

Response 3:

We thank the reviewers for their careful review. The image layout was not adjusted properly. We adjusted and revised it; please see revised Figures 1I, 1J, and Page 7.

Comments 4: In figure 1, E and G should be written in bold. It is the same for figure 2 and the other figures;

Response 4:

We noticed that some of the figure notes were not bolded, so we have made all of the figures bold. Please see the revised manuscript.

Comments 5: - Could it be possible to add uncropped blots as additional files?

Response 5:

Thanks to the reviewers' questions, some of the blots are uncropped blots with markers (see Supplementary Figure), while some of our blots are cropped. The authors did the experiment for the first time and were not aware of the need to keep the original uncropped blots. If necessary, contact us, and we will re-run the western blot experiments and upload the results. Thanks to the reviewer for the reminder.

Comments 6: - The authors reported so many Western blots but for all of them there is no densitometric analysis. The latter is strictly necessary to calculate the difference in expression level between multiple conditions. Could the authors perform densitometric analysis? How many replicates were performed?

Response 6:

      Based on the reviewer's comments, we performed densitometric analysis of western blots for three replicates. Putting the densitometric analysis into each image would have resulted in excessive image layout adjustments, so we placed the results in the supplementary figure.

Comments 7: - To make the reading clearer and simpler, I would suggest reporting more experimental information in the results.

Response 7:

Thanks again to the reviewers for their suggestions. We have added some experiment information to the results to make reading clearer and easier. See the revised manuscript (page 8, line 324; page 11, line 380, line 385, line 391; page 13, line 426, line 428; page 14, line 460; page 16, line 525, line 530; page 18, line 569) for details.

4. Response to Comments on the Quality of English Language

Point 1: Quality of English Language (x) The quality of English does not limit my understanding of the research

Response 1:

      We thank the reviewers for their recognition, and our manuscript was edited and polished by an English-speaking professional organization.

Round 2

Reviewer 2 Report

Comments and Suggestions for Authors

The authors addressed most of the concerns yet there are still some points need to be cleared. 

1. Raw western blottings in Figure 6K and Figure 7P are still concerning. Authors should show uncorpped blots, the assembled blots were not from the same one whole blot.

2. IHC staining in Figure 8D are the same as old version, there is not improvement.

Author Response

Comment 1:Raw western blottings in Figure 6K and Figure 7P are still concerning. Authors should show uncorpped blots, the assembled blots were not from the same one whole blot.

Response 1:Thanks again to the reviewer for your valuable comments on our manuscript. In the past three days, we repeated the western blottings experiments of Figure 6K and Figure 7P, and the results were consistent with the previous experiments. We have placed the new blots in the revised version of the manuscript. We also uploaded the uncropped blots in the supplementary material. Please see the revised Figures 6K and 7P.

Comment 2:IHC staining in Figure 8D are the same as old version, there is not improvement.

Response 2:We have re-selected various groups of other IHC images taken previously for re-loading; please see revised Figure 8.

Reviewer 3 Report

Comments and Suggestions for Authors

Thanks to the authors for replying. However the blots uncropped are not clear as I'm not able to see the marker. Therefore, for future submissions I strongly suggest to provide all blots as it is not adequately correct to submit a paper without any uncropped blot (with this term I mean the raw blot as first raw acquisition)

Author Response

Comment:Thanks to the authors for replying. However the blots uncropped are not clear as I'm not able to see the marker. Therefore, for future submissions I strongly suggest to provide all blots as it is not adequately correct to submit a paper without any uncropped blot (with this term I mean the raw blot as first raw acquisition)

Response:We are grateful to the reviewers for providing valuable comments. The experimental data saved by the students in their experiments were incomplete, and we will follow up to strengthen the experimental data saving, especially to save the uncropped blot data.